# Association of the rs3864283 Polymorphism Located in the *HINT1* Gene with Cigarette Use and Personality Traits

**DOI:** 10.3390/ijms241210244

**Published:** 2023-06-16

**Authors:** Aleksandra Suchanecka, Agnieszka Boroń, Krzysztof Chmielowiec, Aleksandra Strońska-Pluta, Jolanta Masiak, Milena Lachowicz, Jolanta Chmielowiec, Anna Grzywacz

**Affiliations:** 1Independent Laboratory of Health Promotion, Pomeranian Medical University in Szczecin, Powstańców Wielkopolskich 72 St., 70-111 Szczecin, Poland; o.suchanecka@gmail.com (A.S.); aleksandra.stronska@pum.edu.pl (A.S.-P.); 2Department of Clinical and Molecular Biochemistry, Pomeranian Medical University in Szczecin, Aleja Powstańców Wielkopolskich 72 St., 70-111 Szczecin, Poland; agnieszka.boron@pum.edu.pl; 3Department of Hygiene and Epidemiology, Collegium Medicum, University of Zielona Góra, 28 Zyty St., 65-046 Zielona Góra, Poland; chmiele@vp.pl (K.C.); chmiele1@o2.pl (J.C.); 4Second Department of Psychiatry and Psychiatric Rehabilitation, Medical University of Lublin, 1 Głuska St., 20-059 Lublin, Poland; jolanta.masiak@umlub.pl; 5Department of Psychology, Gdansk University of Physical Education and Sport, 80-336 Gdansk, Poland; millkawings@gmail.com

**Keywords:** cigarette use, smoking, personality, anxiety, HINT1

## Abstract

Nicotine is the major reinforcing component of tobacco and it is believed that the pharmacological effects of nicotine motivate the initiation and maintenance of a smoking habit. HINT1 appears to play a role in the modulation of the effects of drug abuse. Hence, the aim of this study was the analysis of the association between the rs3864283 polymorphism of the *HINT1* gene and cigarette use; the analysis of personality traits assessed by the means of the NEO-FFI Inventory; the analysis of anxiety measured by the STAI questionnaire; and the analysis of the interactions between the rs3864283 and both personality traits and anxiety. The study group consisted of 522 volunteers. Of these, 371 were cigarette users and 151 were never-smokers. The genomic DNA was isolated from venous blood using standard procedures. The results of both inventories, i.e., NEO-FFI and STAI., were reported as the sten scores. Genotyping was conducted with the real-time PCR method. Statistically significant differences were found in the frequency of rs3864283 genotypes and alleles in the tested sample of Cigarette Users when compared to the control group. The Cigarette Users compared to the control group obtained higher scores in the assessment of NEO-FFI extraversion scale, and significantly lower results were obtained for the NEO-FFI openness scale, the agreeableness scale, and the conscientiousness scale. There was a statistically significant effect of rs3864283 genotype interaction and Cigarette Use or not using (control group) on the extraversion scale. There was also a statistically significant effect of Cigarette Users or the control group on the extraversion scale score. The results obtained in the presented study indicated a significant association between the *HINT1* rs3864283 variant and smoking status. Moreover, this is the first study incorporating genetic association of above-mentioned polymorphic site with interaction analysis of personality traits and anxiety. Overall, the results of this study suggest that *HINT1* is an important genetic component associated with nicotine usage mechanisms.

## 1. Introduction

Smoking continues to be the leading cause of premature disability and death in the world [1]. Cigarette smoke contains about 7000 different chemicals. At least 70 of these are confirmed or suspected human carcinogens, including arsenic, benzene, formaldehyde, lead, nitrosamines, and polonium 210. The toxic gases carbon monoxide, hydrogen cyanide, butane, toluene, and ammonia are also found in tobacco smoke. Similar toxic substances are emitted by small cigars and hookahs [2].

Nicotine is the major reinforcing component of tobacco and it is believed that the pharmacological effects of nicotine motivate the initiation and maintenance of a smoking habit [3,4]. Nicotine acts through neuronal nicotinic acetylcholine receptors (nAChRs) to exert its pharmacological effects in the brain [5].

The five-factor model of personality [6] has been proposed to classify people with addictive tendencies. This model describes five basic personality dimensions: Neuroticism, Conscientiousness, Extraversion, Agreeableness, and Openness to Experience [6] encompasses an individual’s behavioral, emotional, and cognitive patterns. Investigators using this model in addiction research generally report higher neuroticism in substance dependent individuals compared to controls for both substance use [7,8] and behavioral addictions [9,10]. Such individuals may engage in ‘self-medication’ to alleviate unpleasant or negative emotional states, characteristic of highly neurotic individuals [11]. Another dimension that has yielded relatively consistent results is conscientiousness, which is strongly associated with self-discipline, which is typically lacking in addicts [12]. Lower levels of conscientiousness have been reported in substance abusers [8,13], although more so in drug addicts than in alcohol addicts [14]. Lower levels of Conscientiousness have also been found in behavioral addictions such as gambling disorder (GD) and internet addiction (IA) [9,10,13,15]. Results for agreeableness, extraversion, and openness to experience are less consistent. They also seem to vary across addictions and substance types. Of these dimensions, only low agreeableness was associated with substance use, both alcohol and drugs [16]. Other studies have not found differences in agreeableness, but have found lower levels of extraversion in drug use disorders (DUD) [17]. Several studies have shown higher openness to experience in DUD [7,8], but these were conducted in non-clinical populations. In study Hwang et al. [13], low levels of agreeableness, extraversion, and openness to experience were associated with alcohol symptoms, while a link between higher extraversion and lower openness to alcohol experiences and symptoms was found in another non-clinical study [14].

There have been several models for the classification of temperament and personality. One of the most widely accepted is that of Cloninger, who proposed that there are three genetically homogeneous and independent dimensions of personality: novelty seeking; harm avoidance; and reward dependence [18]. Novelty seeking is the tendency to respond intensely to novel stimuli or cues of potential reward or punishment facilitation, thereby activating/initiating behavior. Harm avoidance is the tendency to respond intensely to aversive stimuli, thereby inhibiting behavior. Reward dependence is the tendency to respond intensely to signals of reward, especially social rewards, thereby maintaining and continuing certain types of behavior. It has been suggested that three dispositions are related to the neurotransmitter system in animal and human brains: novelty seeking would primarily use dopamine pathways, avoidance would use serotonin pathways, and reward dependence would use norepinephrine pathways [19]. Cloninger went on to develop his original Tri-dimensional Personality Questionnaire (TPQ) into a seven-factor model of personality. He developed a new questionnaire called the Temperament and Character Inventory (TCI) [19,20]. The TCI assesses four dimensions of temperament: Harm Avoidance, Novelty Seeking, Reward Dependence, and Persistence, and three character dimensions: Self-Directedness, Cooperativeness, and Self-transcendence.

We decided to utilise the NEO-FFI questionnaire due to a body of evidence suggesting that the NEO-FFI scores in the Polish population seem to correspond better with the assumption about stability of human personality dimensions and genetic factors influencing them [21].

*HINT1*, located on chromosome 5q31.22 in a region involved in linkage and association studies of schizophrenia [22,23,24], encodes a ubiquitously expressed homodimeric purine phosphoramidase of 126 amino acids called Histidine Triad Nucleotide Binding Protein 1. It is one of three HINT proteins (HINT1, HINT2, and HINT3) in the human genome. HINT1 is involved in transcriptional and cell cycle regulation [25]. Little is known about the physiological function of HINT1 protein, although it is widely expressed in the liver, kidney, and brain, including mesocortical and mesostriatal regions [26]. Microarray analysis has identified the gene as a candidate gene involved in the neuropathology of schizophrenia [27,28]. Expression studies have shown that the level of HINT1 mRNA is significantly reduced in the prefrontal cortex (PFc) of male patients with schizophrenia compared to control subjects [29].

It is interesting to note that HINT1 also appears to play a role in the modulation of the effects of drug abuse. Studies investigating the CNS function of HINT1 have shown that the protein specifically interacts with the C-terminus of the μ-opioid receptor. This leads to attenuation of receptor desensitization and inhibition of PKC-mediated μ-opioid receptor phosphorylation [30]. In addition, compared to their wild-type counterparts, *HINT1* knockout mice have enhanced basal and morphine-induced antinociception and improved morphine tolerance [30]. *HINT1* knockout mice are also hypersensitive to the locomotor activating effects of amphetamine and the dopamine receptor agonist apomorphine. This suggests that the absence of *HINT1* is associated with a dysregulation of postsynaptic dopamine transmission [31].

For our analysis, we chose the single nucleotide polymorphism rs3864283 located in the 3′UTR region of the *HINT1* gene. The body of evidence regarding above-mentioned genetic variant is relatively small. rs3864283 is associated with schizophrenia in female Chinese subjects [32]. Study by Jackson et al. [33] reveal the association of the SNP with nicotine dependence. Additionally, human post-mortem mRNA expression showed that smoking status and genotype influence *HINT1* expression in the brain. In animal studies, analyzes showed an increase in *HINT1* protein level in the mouse nucleus accumbens (NAc) after chronic nicotine exposure. Interestingly, after treatment with antagonist of the nicotinic acetylcholine receptors the HINT1 level decreased.

Therefore, the aim of this study was the analysis of the association between the rs3864283 polymorphism of the *HINT1* gene and cigarette use; the analysis of personality traits assessed by the means of the NEO-FFI Inventory; the analysis of anxiety measured by the STAI questionnaire; and the analysis of the interactions between the rs3864283 and both personality traits and anxiety.

## 2. Results

These frequency distributions accorded with the Hardy–Weinberg equilibrium (HWE) both in the Cigarette Users and control subjects (Table 1).

Statistically significant differences were found in the frequency of rs3864283 genotypes in the tested sample of Cigarette Users when compared to the control group (G/G 0.05 vs. G/G 0.11; A/A 0.55 vs. A/A 0.45; A/G 0.40 vs. A/G 0.44, χ^2^ = 7.195, *p* = 0.0274).

Statistically significant differences in the frequency of rs3864283 alleles were found between Cigarette Users and the control group (G 0.25 vs. G 0.33; A 0.75 vs. A 0.67, χ^2^ = 6.200, *p* = 0.0128) (Table 2).

The means and standard deviations of the NEO-FFI results and the STAI state and trait scale for the Cigarette users and control subjects are shown in Table 3.

The Cigarette Users compared to the control group obtained higher scores in the assessment of NEO-FFI extraversion scale (5.96 vs. 5.25; Z = 3.273; *p* = 0.0011). Significantly lower results were obtained for the NEO-FFI openness scale (5.20 vs. 5.69; Z = −2.750; *p* = 0.0060), the agreeableness scale (5.25 vs. 6.35; Z = −4.718; *p* ≤ 0.000), and the conscientiousness scale (5.82 vs. 6.76; Z = −4.251; *p* ≤ 0.000).

The results of the 2 × 3 factorial ANOVA of the NEO Five-Factor Personality Inventory and the State-Trait Anxiety Inventory sten scales are summarized in Table 4.

Significant statistical impact of Cigarette Users and rs3864283 genotype rs3864283 was demonstrated for score of the NEO-FFI Extraversion scale.

There was a statistically significant effect of rs3864283 genotype interaction and Cigarette Use or not using (control group) on the extraversion scale (F_2.516_ = 4.73; *p* = 0.0092; η^2^ = 0.018; Figure 1). The power observed for this factor was 79%, and approximately 2% was explained by the polymorphism of the rs3864283 and Cigarette Users or lack thereof on trait extraversion score variance. There was also a statistically significant effect of Cigarette Users or the control group on the extraversion scale score (F_1.516_ = 11.93; *p =* 0.0006; η^2^ = 0.023). The power observed for this factor was over 93%, and approximately 2% was explained by Cigarette Users or lack thereof on the variance in the extraversion score. In Table 5 are shown the results of the post hoc test.

## 3. Discussion

The aim of our study was the analysis of the association between the rs3864283 polymorphism of the *HINT1* gene and cigarette use; the analysis of personality traits assessed by the means of the NEO-FFI Inventory; the analysis of anxiety measured by the STAI questionnaire; and lastly the analysis of the interactions between the rs3864283 and both personality traits and anxiety.

The results of our association study show statistically significant differences in the frequency of rs3864283 genotypes and alleles in the cigarette users compared to the control group. Similar results were obtained by Jackson, et al. [33]. They showed that two markers in *HINT1* gene (rs3864283 and rs2526303) were associated with nicotine dependency and Fagerström test scores in association studies using VA twins and GAIN controls. The main effects of genotype indicate that individuals with a particular variant found to show association with Fagerström Test for Nicotine Dependence (FTND) score and number of cigarettes they smoke daily, and a significantly higher levels of *HINT1* expression, while the genotype (rs3864283) × smoking status interaction suggests that this effect depends on smoking status, that is, smokers with the rs3864283 risk allele have higher *HINT1* expression than never-smokers. Since the change in mRNA expression in smokers suggests that the polymorphism may have some biological significance in the development of ND, these results support the hypothesis that *HINT1* variants are associated with ND mechanisms. Fang et al. [34] investigated associations between smoking behavior and genetic variants of MOR and MOR-interacting protein genes, including the mu opioid receptor gene *OPRM1* and two mu opioid receptor-interacting protein genes *ARRB2* and *HINT1*, they conducted a cross-sectional study among Chinese male smokers. *HINT1* rs3852209 was significantly associated with smoking status (current smokers vs. former smokers) in Chinese men with tobacco-related diseases.

The results of the NEO-FFI Inventory analysis showed that cigarette users had significantly higher scores on the NEO-FFI extraversion scale, while simultaneously having statistically significantly lower scores on the openness scale, agreeableness scale, and conscientiousness scale. In our study, we also found significant statistical interaction of Cigarette Users and genotype rs3864283, which was demonstrated to impact the score of the Extraversion scale results of NEO-FFI. Waga et al. [35] carried out the NEO-FFI psychological test with analysis of the CYP2A6 gene to elucidate the mechanism of formation of an individual’s smoking behavior in relation to their personality and temperament. They found no significant differences in Neuroticism, Extraversion, Agreeableness, and Conscientiousness scores according to the NEO-FFI between smokers and non-smokers. Kulkarni et al. [36] conducted a study to assess the level of nicotine dependence in tobacco smokers (working in the corporate sector) to find out their personality profile and the association of their personality traits with continued smoking. They showed that Neuroticism was significantly related to the level of nicotine dependence. Extraversion and openness were related to health concerns, while agreeableness and conscientiousness were related to social factors as a reason for quitting smoking. Extraversion and agreeableness were associated with occupational and social factors as reasons for relapse.

We found a statistically significant interaction effect of rs3864283 genotype and smoking status (cigarette use or control group) on the extraversion scale. The observed potential for this factor was 79%, and approximately 2% was explained by the rs3864283 and cigarette user polymorphism or lack thereof on the variance of the trait extraversion score. A statistically significant effect of cigarette smoker or control group on the extraversion scale score was also observed. The observed potential for this factor was more than 93%, and about 2% was explained by cigarette users or lack thereof on the variance of the extraversion score.

Jackson, et al. [33] also examined protein expression in vivo and the results showed that HINT1 protein levels did not change in any brain area tested after a single nicotine injection; however, there was a significant increase in protein levels in the NAc after chronic nicotine exposure, and this increase was reduced after a single injection of the non-selective nAChR antagonist, mecamylamine, suggesting that the chronic nicotine-induced increase in the HINT1 level is mediated directly through nAChRs. Furthermore, the nicotine-induced changes in the HINT1 level were only observed in the NAc, indicating a brain-region-specific effect. Indeed, the NAc is a brain region implicated in ND behaviors, including nicotine reward [37,38,39], self-administration [40], and withdrawal [27,30]. On the basis of these observations, it can be stated that the nicotine-induced increase of HINT1 is mediated through specific nAChR subtypes, such as α4β2* (where * denotes the possible incorporation of additional subunits) or α7, the major nAChR subtypes in the brain, which have been shown to mediate behaviors associated with ND. It cannot be ruled out that the changes observed in the HINT1 level specifically in the NAc after cessation of nicotine treatment have some relevance in ND. In another study, Jackson, et al. [41] attempted to determine the behavioral role of HINT1 in nicotine dependence. To do so, they tested male *HINT1* wild-type (+/+) and knockout (−/−) mice in a nicotine reward test (CPP), a nicotine withdrawal model assessing both physical and affective symptoms, and a conditioned place aversion (CPA) test of nicotine withdrawal. *HINT1* −/− mice did not develop a significant CPP of nicotine, and physical withdrawal symptoms (hyperalgesia and somatic symptoms) were attenuated in *HINT1* −/− mice. Conversely, *HINT1* −/− mice developed a significant CPA of nicotine withdrawal similar to their ++ counterparts.

Liu et al. [42] conducted a study in which they extensively investigated HINT1 protein involvement in key brain areas associated with addiction, including prefrontal cortex, nucleus accumbens, striatum, and hippocampus at different stages of different models. They also investigated the effect of HINT1 protein deletion on morphine addiction using *HINT1* knockout mice to establish the above models and a physical dependence model. They found that in many animal models of addiction, HINT1 is involved to varying degrees at different stages. The absence of HINT1 may have some attenuating effect on morphine-mediated addiction behavior and may alleviate morphine withdrawal symptoms.

Our study is not free of some limitations. We analyzed only people of Caucasian origin; hence, the results ought to be verified in other populations. Additionally, we analyzed only one SNP in the *HINT1* gene, therefore our reasoning capabilities based on this amount of data are restricted. Our future analysis will incorporate methylation analysis on greater number of subjects and also subjects addicted to different substances to further analyze the role of *HINT1* gene in greater detail.

## 4. Materials and Methods

### 4.1. Participants

The study group consisted of 522 volunteers. Of these, 371 were cigarette users (mean age = 29.44, SD = 10.74; F = 49%, M = 51%) and 151 were never-smokers (mean age = 26.91, SD = 10.10; F = 80%, M = 20%). The Bioethics Committee of the Pomeranian Medical University in Szczecin approved the study (KB-0012/164/17-A). All participants gave their written, informed consent prior to entering the study. The study was conducted in the Independent Health Promotion Laboratory, Pomeranian Medical in Szczecin. Both cigarette users and the control group underwent the same psychometric testing with the NEO Five-Factor Personality Inventory (NEO-FFI) and State-Trait Anxiety Inventory (STAI) questionnaire.

### 4.2. Psychometric Tests

STAI questionnaire measures anxiety as a trait which can be described as a persistent predisposition to having worries, stress, discomfort, and anxiety as a state, which can be described as anxiety, fear, and momentary stimulation of the autonomic nervous system in response to specific situations.

The Personality Inventory (NEO-FFI Five-Factor Inventory, NEO-FFI) contains six components for each of the five traits-neuroticism (anxiety, hostility, depression, self-awareness, impulsivity, and susceptibility to stress), extroversion (warmth, sociability, assertiveness, activity, emotion seeking, and positive emotions), openness to experience (fantasy, aesthetics, feelings, actions, ideas, and values), agreeableness (trust, straightforwardness, altruism, compliance, modesty, and tenderness), and conscientiousness (competence, order, duty, striving for achievements, self-discipline, and consideration) [43].

The results of both inventories, i.e., NEO-FFI and STAI., were reported as the sten scores. The conversion of the raw score to the sten scale was carried out following the Polish standards for adults, where it was assumed that 1–2 sten corresponds to very low results; 3–4 low results, 5–6 average results; 7–8 high results, and 9–10 sten corresponds to very high results.

### 4.3. Genotyping

The genomic DNA was isolated from venous blood by using standard procedures. Genotyping was conducted with the real-time PCR method. The fluorescence signal was plotted as a function of temperature to provide melting curves for each sample. The *HINT1* gene rs3864283 polymorphic site peaks were read at 53.29 °C for the A allele and at 59.93 °C for the G allele.

### 4.4. Statistical Analysis

A concordance between the genotype frequency distribution and Hardy–Weinberg equilibrium (HWE) was tested using the HWE software (https://wpcalc.com/en/equilibrium-hardy-weinberg/ accessed on 5 April 2023). The relations between rs3864283 variants: cigarette users and control subjects and the NEO Five-Factor Inventory were analyzed using a multivariate analysis of factor effects ANOVA [NEO-FFI/scale STAI/× genetic feature × control and cigarette users × (genetic feature × control and cigarette users)]. The condition of homogeneity of variance was fulfilled (Levene test *p* > 0.05). The analyzed variables were not distributed normally. The NEO Five-Factor Inventory (Neuroticism, Extraversion, Openness, Agreeability, and Conscientiousness) sten scores were compared using the U Mann–Whitney test. rs3864283 genotype frequencies between control subjects and cigarette users were tested using the chi-square test. All computations were performed using STATISTICA 13 (Tibco Software Inc, Palo Alto, CA, USA) for Windows (Microsoft Corporation, Redmond, WA, USA).

## 5. Conclusions

The results obtained in the presented study indicated a significant association between the *HINT1* rs3864283 variant and smoking status. Moreover, this is the first study incorporating genetic association of above-mentioned polymorphic site with interaction analysis of personality traits and anxiety. Overall, the results of the present study suggest that *HINT1* gene is a genetic component associated with nicotine usage.

## Figures and Tables

**Figure 1 ijms-24-10244-f001:**
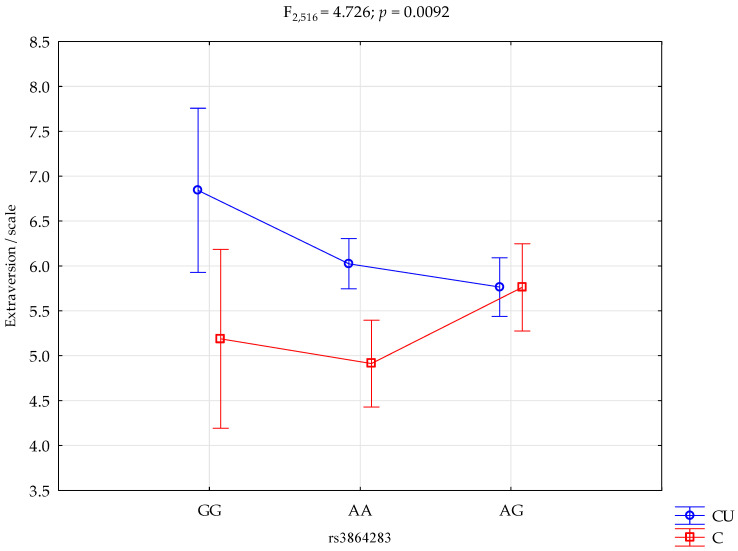
Interaction between Cigarette Users (CU)/control (C) and rs3864283 and Extraversion scale.

**Table 1 ijms-24-10244-t001:** Hardy–Weinberg’s equilibrium for rs3864283 located in the *HINT1* gene.

Hardy-Weinberg Equilibrium Calculator Including Analysis for Ascertainment Bias		Observed (Expected)	Allele Freq	χ^2^ (*p* Value)
rs3864283 Cigarette Users *n* = 371	G/G	19 (23.6)	*p* (ins) = 0.75 q (del) = 0.25	1.580 (0.2088)
	A/A	203 (207.6)		
	A/G	149 (139.9)		
rs3864283 control *n* = 151	G/G	16 (16.2)	*p* (ins) = 0.67 q (del) = 0.33	0.0070 (0.9332)
	A/A	68 (68.2)		
	A/G	67 (66.5)		

*p*–statistical significance χ^2^ test.

**Table 2 ijms-24-10244-t002:** Frequency of genotypes of the rs3864283 polymorphisms in *HINT1* gene in the Cigarette Users and control subjects.

	rs3864283				
	Genotypes			Alleles	
	G/G n(%)	A/A n(%)	A/G n(%)	G n(%)	A n(%)
Cigarette Users *n* = 371	19 (5.12%)	203 (54.72%)	149 (40.16%)	187 (25.20%)	555 (74.80%)
Control *n* = 151	16 (10.60%)	68 (45.03%)	67 (44.37%)	99 (32.78%)	203 (67.22%)
χ^2^ (*p* value)	**7.195** **(0.0274) ***			**6.200** **(0.0128) ***	

*n*—number of subjects. */bold—significant statistical differences.

**Table 3 ijms-24-10244-t003:** STAI and NEO Five-Factor Inventory sten scores between healthy controls and Cigarette Users.

STAI/NEO Five-Factor Inventory/	Cigarette Users (*n* = 371)	Control (*n* = 151)	Z	(*p*-Value)
STAI trait/scale	5.95 ± 2.22	5.64 ± 2.18	1.843	0.0652
STAI state/scale	5.56 ± 2.35	5.50 ± 2.23	0.562	0.5741
Neuroticism/scale	5.93 ± 2.22	5.71 ± 1.99	1.460	0.1442
Extraversion/scale	5.96 ± 2.09	5.25 ± 1.96	3.273	**0.0011 ***
Openness/scale	5.20 ± 2.03	5.69 ± 2.01	−2.750	**0.0060 ***
Agreeability/scale	5.25 ± 2.25	6.35 ± 2.37	−4.718	**0.0000 ***
Conscientiousness/scale	5.82 ± 2.13	6.76 ± 2.25	−4.251	**0.0000 ***

*p*, statistical significance with Mann–Whitney U-test; *n*—number of subjects; M ± SD, mean ± standard deviation; */bold statistically significant differences.

**Table 4 ijms-24-10244-t004:** The results of 2 × 3 factorial ANOVA for cigarette users and controls, NEO Five Factor Inventory, STAI, and HINT1 rs3864283.

STAI/NEO Five-Factor Inventory	Group	rs3864283				ANOVA		
		G/G *n* = 35 M ± SD	A/A N *=* 271 M ± SD	A/G *n* = 216 M ± SD	Factor	F (*p* Value)	ɳ^2^	Power (alfa = 0.05)
STAI trait scale	Cigarette Users (CU); *n* = 371	5.53 ± 2.84	6.16 ± 2.50	5.73 ± 2.25	intercept CU/control rs3864283 CU/control x rs3864283	F_1,516_ = 1301.26 (*p* < 0.0001) F_1,516_ = 1.88 (*p* = 0.1710) F_2,516_ = 2.62 (*p* = 0.074) F_2,516_ = 0.49 (*p* = 0.6098)	0.718 0.004 0.010 0.002	1.000 0.277 0.521 0.131
	Control; *n* = 151	4.63 ± 1.96	5.85 ± 2.07	5.67 ± 2.28				
STAI state scale	Cigarette Users (CU); *n* = 371	5.73 ± 2.47	5.70 ± 2.44	5.33 ± 2.21	intercept CU/control rs3864283 CU/control x rs3864283	F_1,516_ = 1284.59 (*p* < 0.0001) F_1,516_ = 0.594 (*p* = 0.4411) F_2,516_ = 0.34 (*p* = 0.7113) F_2,516_ = 0.96 (*p* = 0.3829)	0.713 0.001 0.001 0.004	1.000 0.120 0.104 0.217
	C: Control; *n* = 151	4.94 ± 2.32	5.53 ± 2.22	5.60 ± 221				
Neuroticism scale	Cigarette Users (CU); *n* = 371	5.26 ± 2.84	5.89 ± 2.26	6.07 ± 2.07	intercept CU/control rs3864283 CU/control x rs3864283	F_1,516_ = 1596.18 (*p* < 0.0001) F_1,516_ = 0.19 (*p* = 0.6586) F_2,516_ = 1.59 (*p* = 0.2046) F_2,516_ = 0.07 (*p* = 0.9352)	0.756 0.0004 0.006 0.0003	1.000 0.073 0.337 0.060
	Control; *n* = 151	5.31 ± 1.74	5.65 ± 1.84	5.88 ± 2.17				
Extraversion scale	Cigarette Users (CU); *n* = 371	6.84 ± 1.54	6.02 ± 2.17	5.76 ± 2.00	intercept CU/control rs3864283 CU/control x rs3864283	F_1,516_ = 1847.76 (*p* < 0.0001) **F_1,516_ = 11.93 (*p* = 0.0006) *** F_2,516_ = 1.66 (*p* = 0.1904) **F_2,516_ = 4.73 (*p* = 0.0092) ***	0.782 0.023 0.006 0.018	1.000 0.931 0.351 0.789
	Control; *n* = 151	5.19 ± 1.64	4.91 ± 2.03	5.76 ± 1.80				
Openness scale	Cigarette Users (CU); *n* = 371	5.21 ± 2.04	5.19 ± 1.91	5.21 ± 2.18	intercept CU/control rs3864283 CU/control x rs3864283	F_1,516_ = 1653.66 (*p* < 0.0001) F_1,516_ = 2.96 (*p* = 0.0858) F_2,516_ = 0.37 (*p* = 0.6931) F_2,516_ = 0.31 (*p* = 0.7367)	0.762 0.006 0.001 0.001	1.000 0.404 0.109 0.099
	Control; *n* = 151	5.62 ± 2.53	5.51 ± 1.95	5.85 ± 1.95				
Agreeability scale	Cigarette Users (CU); *n* = 371	5.53 ± 2.84	5.15 ± 2.20	5.36 ± 2.24	intercept CU/control rs3864283 CU/control x rs3864283	F_1,516_ = 1482.32 (*p* < 0.0001) **F_1,516_ = 8.98 (*p* = 0.0028) *** F_2,516_= 1.12 (*p* = 0.3256) F_2,516_ = 0.47 (*p* = 0.6234)	0.741 0.017 0.004 0.002	1.000 0.848 0.248 0.127
	Control; *n* = 151	6.00 ± 2.42	6.13 ± 2.30	6.61 ± 2.42				
Conscientiousness scale	Cigarette Users (CU); *n* = 371	5.58 ± 2.19	5.94 ± 2.21	5.72 ± 1.98	intercept CU/control rs3864283 CU/control x rs3864283	F_1,516_ = 1896.27 (*p* < 0.0001) **F_1,516_ = 10.72 (*p* = 0.0011) *** F_2,516_ = 0.32 (*p* = 0.7260) F_2,516_ = 0.18 (*p* = 0.8352)	0.786 0.020 0.001 0.0007	1.000 0.904 0.101 0.078
	Control; *n* = 151	6.50 ± 1.93	6.75 ± 2.18	6.79 ± 2.43				

*/bold—significant result; CU—Cigarette Users; M ± SD-mean ± standard deviation.

**Table 5 ijms-24-10244-t005:** Post hoc test (Bonferroni) analysis of interactions between the patients diagnosed with Cigarette Users/control and rs3864283 and Extraversion scale.

rs3864283 and Extraversion Scale						
	{1} M = 6.84	{2} M = 6.02	{3} M = 5.76	{4} M = 5.19	{5} M = 4.91	{6} M = 5.76
Cigarette Users G/G {1}		0.0937	0.0298	0.0166	**0.0003 ***	0.0409
Cigarette Users A/A {2}			0.2363	0.1127	**0.0001 ***	0.3572
Cigarette Users A/G {3}				0.2797	0.0042	0.9896
Control G/G {4}					0.6249	0.3100
Control A/A {5}						0.0153
Control A/G {6}						

*/bold—significant statistical differences, M—mean.

## Data Availability

Not applicable.

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
