# Peer review of "Association of the rs3864283 Polymorphism Located in the HINT1 Gene with Cigarette Use and Personality Traits"

_ijms, 2023, doi:10.3390/ijms241210244_

Round 1
Reviewer 1 Report
The topic is very actual, the study is methodologically sound.
I have only one suggestion and one question ad Methods
1. Why did you choose the Personality Inventory (NEOFFI) ? Could you briefly discuss in this connection Cloninger's biosocial model of personality?
2. I suggest adding a brief comment (into Discussion and/or Conclusions) on importance of you results for clinical practice and a vision for future (personalized medicine, changing the concept of health care preventive, not reactive).
Author Response
Dear Reviewer,
We would like to thank you for your valuable comments on the article. Below you will find our reply to your review. All changes are with a description or a comment and changes have been made to the manuscript (track changes in the tracking group on the review tab).
Reviewer 1
The topic is very actual, the study is methodologically sound.
I have only one suggestion and one question ad Methods
- Why did you choose the Personality Inventory (NEOFFI)? Could you briefly discuss in this connection Cloninger's biosocial model of personality?
Thank you for this comment – the paragraph has been added – starting from line 78.
- I suggest adding a brief comment (into Discussion and/or Conclusions) on importance of you results for clinical practice and a vision for future (personalized medicine, changing the concept of health care preventive, not reactive).
Thank you for this comment – the information has been added – starting from line 321.
In addition, during the final proofreading, we noticed that the rs number was entered incorrectly, so in the entire manuscript, the rs number has been corrected.

Reviewer 2 Report
This manuscript deals with the association between a genetic polymorphism of the HINT1 gene, cigarette use, and personality traits. It is a fairly straightforward genetic polymorphism association study including just one such polymorphism (rs3864236). The manuscript is solidly written and that is why I struggle to find a reason for not including at least a hint of logic behind selecting this gene and the said gp in the introduction. The authors mention the association between this gene and schizophrenia, as well as the modulation of the effects of drug abuse. None of them have that much with acetylcholine (smoking is the theme of the paper and as far as I gathered the authors did not include patients with psychiatric diagnoses, save maybe nicotine dependence). However, they somewhat corrected that issue in the discussion. Having said that, they do not mention the rs3864236, but HINT1 protein. We do not go into certain details about this gp, is it a functional polymorphism? How did the authors decide on this and not some other gp, snp, indel and so on? Why did they not include several of them (the subject number seems appropriate for that). Because of all laid out it seems that the said gp was selected randomly. Overall, a lot of discussion section actually belongs to the introduction. The manuscript lacks a limitation section, although I mentioned a few of them above. The conclusion is a bold one, the association may be significant, but suggesting that HINT1 is an important genetic component is an overstatement. The authors did not investigate HINT1, just one of its gp's!
Strong points in my opinion are the subject size number (well over 500) for a genetic association analysis, HWE met, and sound statistics with a moderately conservative Bonferroni correction.
Also, NEO-FFI and STAI are valid instruments, a bit overused in my opinion, but still valid.
I believe this is solid research although gp analyses (especially single gp) are a bit passe.
Author Response
Dear Reviewer,
We would like to thank you for your valuable comments on the article. Below you will find our reply to your review. All changes are with a description or a comment and changes have been made to the manuscript (track changes in the tracking group on the review tab).
Reviewer 2
This manuscript deals with the association between a genetic polymorphism of the HINT1 gene, cigarette use, and personality traits. It is a fairly straightforward genetic polymorphism association study including just one such polymorphism (rs3864236). The manuscript is solidly written and that is why I struggle to find a reason for not including at least a hint of logic behind selecting this gene and the said gp in the introduction. The authors mention the association between this gene and schizophrenia, as well as the modulation of the effects of drug abuse. None of them have that much with acetylcholine (smoking is the theme of the paper and as far as I gathered the authors did not include patients with psychiatric diagnoses, save maybe nicotine dependence). However, they somewhat corrected that issue in the discussion. Having said that, they do not mention the rs3864236, but HINT1 protein. We do not go into certain details about this gp, is it a functional polymorphism? How did the authors decide on this and not some other gp, snp, indel and so on? Why did they not include several of them (the subject number seems appropriate for that). Because of all laid out it seems that the said gp was selected randomly. Overall, a lot of discussion section actually belongs to the introduction. The manuscript lacks a limitation section, although I mentioned a few of them above. The conclusion is a bold one, the association may be significant, but suggesting that HINT1 is an important genetic component is an overstatement. The authors did not investigate HINT1, just one of its gp's! Strong points in my opinion are the subject size number (well over 500) for a genetic association analysis, HWE met, and sound statistics with a moderately conservative Bonferroni correction. Also, NEO-FFI and STAI are valid instruments, a bit overused in my opinion, but still valid. I believe this is solid research although gp analyses (especially single gp) are a bit passe.
Thank you for this suggestions, appropriate changes were made in the manuscript:
– the description of rs3864236 has been added – starting from line 120
– the limitations section was added – starting from line 267
– after studying the data once more we concluded that our conclusions might not fit perfectly the results, and we changed this part of paper – starting from line 321.
In addition, during the final proofreading, we noticed that the rs number was entered incorrectly, so in the entire manuscript, the rs number has been corrected.
